# Analysis of Enterprise Sustainable Crowdsourcing Incentive Mechanism Based on Principal-Agent Model

**Guohao Wang** and **Liying Yu** *

School of Management, Shanghai University, Shanghai 200444, China
* Correspondence: yuliying@shu.edu.cn; Tel.: +86-021-6613-3851

**Abstract:** The utilization of crowdsourcing to acquire distant knowledge is increasing. In the new product development process, sustainable crowdsourcing is an effective way to exploit both external and internal resources to boost enterprise innovation quality and the efficiency of the competitive edge of macro tasks in a relatively long cycle. The challenge of sustainable crowdsourcing is how to design a proper incentive mechanism to achieve the maximum initiator profit and, at the same time, satisfy the solver's motivation so that they can continuously participate in the innovation process. In two situations, including a single motivation and multiple motivations of the solver, this paper analyzed the impact of a few factors on the initiator's profit and the incentive coefficient for the solver based on the Principal–Agent Model. From the model and simulation results, the solver's incentive coefficient is positively correlated to the solver's work quality and negatively correlated to the uncertainty of the enterprise operation, the solver's Effort Cost, the solver's degree of risk aversion, etc. If the initiator is more sensitive to the benefits of the solver's intrinsic motivation, the monetary incentive will be higher. The research results provide a theoretical basis to quantify the initiator's expected profit and design a proper incentive plan for the solver. Finally, the conclusions offer practical guidance for enterprise to execute incentive plans for sustainable crowdsourcing from the perspective of the solver's motivation.

**Keywords:** sustainable crowdsourcing; incentive; solver's motivation; new product development

## 1. Introduction

With the development of information technology, an enterprise can make full use of the creative potential of both the internal and external knowledge of the huge online population by establishing a distributed work model. Today, crowdsourcing innovation across business areas can solve certain problems faster, better, and more cheaply than other traditional solutions within a company [1–3]. During crowdsourcing activity, the enterprise can gather social resources, partners, clients and internal employees to transfer knowledge and drive the organization to sustainably innovate.

More and more enterprises, especially high-tech companies, such as Xiaomi, Systems, Applications and Products (SAP), Google, Dell, General Electric (GE), Procter & Gamble Company (P & G), Lego, etc., are starting to build their own crowdsourcing platforms to place the heavy responsibility of creating core corporate values on the internal and external users. GE has established an enterprise crowdsourcing activity called "Ecomagination Challenge" [4], which aims to collect novel technological ideas from the general public. General Electric collected 5,000 ideas from different participants in 160 countries through a crowdsourcing platform, of which 23 ideas were extremely valuable and were eventually absorbed into the company's product line. The "Dell IdeaStorm" [5] online crowdsourcing community was established in 2007, with the main purpose of interacting with employees, partners

and clients in crowdsourcing activities to improve the quality of product feedback and new products. Google Translate [6] is based on web 2.0 technology, using a crowdsourcing mode to collect feedback from users around the world on the quality of translation in multiple languages, combined with machine learning algorithms to continuously improve the accuracy of translation, and achieve the self-learning of translation evolution.

For macro task crowdsourcing activities, e.g., business to business (B2B) crowdsourcing [7–9], for science and technology, which usually include a few subtasks, enterprise delegates product designs, research and development, or key technologies to different communities, allowing the participants to discover ideas or solve technical issues, or introduce collected user feedback into the product development process to improve the efficiency and quality of product development. In order to solve the problems, the enterprise, as an initiator in crowdsourcing, expect that the solvers are willing to participate and work hard to complete the task. Moreover, for some specific and professional tasks, the initiator hopes that the solvers will participate multiple times. In this study, we introduce a new type of concept—sustainable crowdsourcing. Instead of a one-time solution, sustainable crowdsourcing process operates in a relatively long cycle, usually with multiple iterations between initiator and solver towards subtask, which require the solver's continuous participation to maintain output at certain rate or level. Taking New Product Development (NPD) of a large B2B company as an example, the full life cycle of product development is generally divided into three stages: Fuzzy Front End (FFE), Development and Commercialization [10,11]. The whole process of crowdsourcing may take a few months or even years, during which it is critical to maintain the high quality of the crowdsourcing subtasks. This is a typical model for sustainable crowdsourcing. The characteristics of the crowdsourcing model, corresponding to the three phases and the solvers, are shown in Table 1.

**Table 1.** Enterprise crowdsourcing characteristics during New Product Development (NDP) phases.

| NPD Phases | Crowdsourcing Characteristics | Solvers | Typical Subtasks |
|---|---|---|---|
| FFE | Mainly from enterprise internal users<br>The target is to identify opportunities and concepts | Employees | Strategy planning, research, idea generation, idea evaluation, business analysis |
| Development | Inputs are mainly from partners or key clients<br>The target is to develop a prototype | Partners, key clients | Prototype design, engineering, technical evaluation, prototype testing |
| Commercialization | Feedback is collected from a wide range of clients<br>The target is to promote the production line, improve product quality and reduce cost | Wide range of clients | Production, commercialization |

By providing knowledge, time, ideas and solutions in crowdsourcing activity, solvers can help the enterprise to resolve diverse and complex problems, so that the enterprise, as the initiator, can obtain knowledge, experience and problem solutions, and eventually boost company business. For enterprise crowdsourcing, solvers can be a large number of knowledgeable, unknown and heterogeneous individual participants, including employees, partners, clients, etc. In order to achieve sustainable crowdsourcing activity, the initiator should design the process properly to maintain a close cooperation between initiator and solver—in particular, an incentive mechanism is critical for healthy and sustainable crowdsourcing. This paper focuses on sustainable crowdsourcing for enterprises and analyzes the solver's incentive mechanism based on the Principal–Agent Model. To probe the incentive mechanism of sustainable crowdsourcing for enterprises, this paper is set out to answer the following research question: *How can the enterprise determine the best incentive model that the initiator can offer to solvers in order to keep the sustainable crowdsourcing process going?*

The main purpose of enterprise sustainable crowdsourcing is to acquire external and internal participants' knowledge for continuous implementation in organizational innovation activities, so that it can achieve mutual penetration, interaction and collaborative innovation between the crowdsourcing initiator and solvers through crowdsourcing platforms. Crowdsourcing solvers often have professional knowledge, novel ideas or creativity, all of which can be regarded as innovative elements for enterprise techniques and business. Therefore, whether crowdsourcing participants can continue to support crowdsourcing activities—that is, the motivation of crowdsourcing solvers—has become one of the keys to the success of sustainable crowdsourcing operations. Our research, based on the Principal–Agent (PAM) model, has practical application scenarios, since there is a contractual relationship between the initiator and solver, and the incentive mechanism is critical for improving sustainable crowdsourcing performance.

The purpose of this research is to explore the incentive mechanism for sustainable crowdsourcing using a quantitative approach. Our research, which is theoretical research on practical problems, presents a novel methodology by establishing a game model to analyze crowdsourcing participants' motivations and the impact on crowdsourcing revenues. The rest of the paper is organized as follows: the recent literature is reviewed in Section 2. Model assumptions, notation, and formulations are developed in Section 3. Numerical simulations and discussions are presented in Section 4. Finally, some conclusive remarks are provided in Section 5.

## 2. Literature Review

In terms of new product development, crowdsourcing can be used for diverse types of subtasks. In this paper, we focus on initiator enterprise soliciting solutions for technology needs. Crowdsourcing for technology needs is often undertaken as part of a regular research and development project [12], addressing concrete development tasks such as new product development. The composition of crowdsourcing participants is different at different stages of enterprise product development [10]. For each stage, there are a few subtasks which are available for crowdsourcing innovation [13,14]. In the Fuzzy Front-End stage of new product development, the crowdsourcing solvers are mainly anonymous internal employees, since internal solvers are more familiar with the organization's objective and these solver's solutions are more feasible, while internal solvers would also feel safe because they are anonymous participants [15]. In the development and commercialization stage of new product development, the solvers are mainly comprised of people outside the enterprise, such as partners, users, etc. As an initiator, the enterprise should design corresponding incentive mechanisms to ensure that different solvers at different stages can continue to participate in the corporate innovation process [16], so that sustainable crowdsourcing can benefit the enterprise and the business, and ultimately promote corporate innovation.

Attracting solvers' continuous participation has been viewed as an important step to achieve sustainable crowdsourcing in organizations. Zhong et al. [17] found that both distribution fairness and procedural fairness have an impact on continued participation in crowdsourcing. Jin et al. [18] indicated that solvers' satisfaction and affective commitment had an impact on the continuous participation willingness of the online community. The reason may be that the solver can establish and strengthen their self-assessment [19] through the iterations of sustainable crowdsourcing. Zheng et al. [20] explained that the monetary incentive is not the only factor to motivate solvers' continuous participation. Feller et al. [21] studied the effect and role of solver brokerages, one type of innovation intermediary, to enable organizations to sustainably acquire distant IP from solvers in three processes, including knowledge mobility, appropriability and stability. In addition, the scientific crowdsourcing platform also has an impact on solvers' sustainable development experience [22].

The proper incentives are critical elements to attract the solver to continuously participate in crowdsourcing activities, while satisfying the solver's versatile motivations [18,20] is a good way to create effective incentives. The motivation of crowdsourcing participants mainly refers to the willingness of the crowdsourcing solvers to transfer their knowledge and information to the

initiator. From the perspective of enterprise product development, the motivations of crowdsourcing participants can be divided into intrinsic motivation and extrinsic motivation [23,24]. Intrinsic motivation mainly refers to the intrinsic self-motivation of participants. According to Maslow's Theory of Self-Actualization, the intrinsic motivation of participants can be separated into two categories: psychological needs and self-improvement. Self-esteem, social network building, gaining extensive attention and entertainment are included in the category of psychological needs, while learning new skills, acquiring new knowledge, winning employment opportunities, etc., fall into the category of self-improvement. Extrinsic motivation mainly refers to rewards, bonuses, gifts, money, etc. The monetary incentive is the most important extrinsic motivation of the solvers to participate in crowdsourcing. During the enterprise product development process, the cooperation of people with different professional backgrounds and knowledge is often required. Participants often want to learn new skills or new knowledge at the same time as getting paid. Therefore, setting up monetary incentives and providing opportunities to learn new knowledge are important approaches for applying sustainable crowdsourcing in enterprise and continuously attracting participants [25–27].

In terms of extrinsic motivation, many scholars believe that the most influential external factor for crowdsourcing is money [28,29]. Monetary incentives can strengthen social participation behaviors [30–32], and monetary incentives include welfare, cash, income and part time jobs [28]. Organisciak [33] conducted research on multiple crowdsourcing websites and found that the main reason for social crowd participation in crowdsourcing is monetary incentives, which are more effective and motivating than all other incentives. Liu et al. [34] concluded that the material rewards or monetary incentives of crowdsourcing can significantly affect the enthusiasm of crowdsourcing participants, and also impact the quality and quantity of crowdsourcing output. Based on the Theory of Planned Behavior (TPB), Han [35] found that the solver's expectation of revenue from crowdsourcing can significantly positively affect their willingness to participate. Boudreau et al. [36] analyzed the users of the Topcoder website and found that the main motivation for the crowdsourcing solver to participate in crowdsourcing competitions is to obtain bonuses and win the attention and internship opportunities of relevant IT companies.

In terms of intrinsic motivation, many studies indicate that most crowdsourcing participants take part for non-remuneration reasons. DiBona et al. [37] researched the open source innovation platform and found that the important reasons for users to participate in open source tasks include gaining the recognition of peers in the industry and showing their skills to obtain jobs or internship opportunities within companies in the industry. Zhong et al. [38] believes that the key to the success of the crowdsourcing community lies in the continuous participation behavior of the subcontractor, and the reasons for the continuous participation of the subcontractor are mainly due to the inherent factors of the subcontractor, such as satisfaction and self-motivation. From the analysis of online crowdsourcing, Ye et al. [39] found that users' mainly participated in crowdsourcing to obtain a sense of joy and happiness. Ipeirotis et al. [40] analyzed the Amazon Mechanical Turk (MTurk) crowdsourcing platform and found that the motivation of most participants came from non-monetary factors, such as entertainment and building social networks. Tran et al. [11] proposed the concept of the "Prosumer", which means that consumers, in order to satisfy their own interests, deeply participate in product design and become producers as well. Zheng et al. [20] argue that intrinsic motivation is more important than extrinsic motivation to encourage crowds to participate. They believed that a balance of both extrinsic and intrinsic motivation is necessary to promote participation in crowdsourcing.

To date, many studies have been performed to analyze the key factors that impact crowdsourcing solvers' motivations. It has been commonly concluded by scholars that motivations can be defined as intrinsic and extrinsic, and both of them impact the participant's willingness and output. The crowdsourcing solver's typical motivations are listed in Table 2. However, as Zhao et al. [41] point out, it is very interesting to explore the relationship between the crowd's effort and the quantity of their contributions, as well as the incentives and expected behaviors. Given the characteristics of sustainable crowdsourcing, these topics are important; however, the relevant research is limited.

Furthermore, current research is more qualitative, such as concept description, motivation introduction and classification, etc. It is worth doing a quantitative analysis to see how enterprises can design proper incentive mechanisms to motive participants to take part in sustainable crowdsourcing.

**Table 2.** Typical crowdsourcing solver's motivation summary.

| Motivation Type | Motivation Category | Typical Cases |
|---|---|---|
| Intrinsic | Obtaining social recognition and reputation | Thingivers, Wikipedia, SAP Community Network |
| | Learning new knowledge and skill | Wikipedia, Google Translate platform |
| | Gaining joy, fun and attention | Dell IdeaStorm, Threadless, MI Community, LEGO Cuusoo platform |
| Extrinsic | Monetary and financial rewards | Innocentive, MTurk, Threadless, Youtube, Flickr, CrowdANALYTIX, Planbox, Taskcn |

Although there are a few studies that use different quantitative approaches, such as decision-making trials and evaluation laboratories with fuzzy set theory [42], empirical research [43] and evolutionary theory [44], to investigate crowdsourcing, these methods are not suitable for the case of the solver's incentive for participating in sustainable crowdsourcing, especially considering the scenarios of the solver's intrinsic and extrinsic motivations, which are the key factors of establish the game relationship between the initiator and solver. Because Principal–Agent theory is often used to guide principals in designing incentive mechanisms that induce agents to act according to the principal's wishes, they are often used in the design of compensation incentive mechanisms [45,46]. Obviously, the relationship between the initiator and the solver is a typical Principal–Agent relationship, in which the initiator is the principal and the solver accepts the task and is the agent. In the design of the solver's incentive mechanism, there is a typical information asymmetry: the solver has knowledge that the initiator does not have, and the principal (initiator) has asymmetrical information about the agent (solver)'s effort level. Moreover, in the process of crowdsourcing cooperation, the solvers, as agents, will adjust their work efforts according to the incentives from the initiator.

## 3. Model Formulation

This paper uses the Principal–Agent Model to study the following two scenarios. (1) Participants are driven by a single motivation, which is a monetary incentive. (2) Participants are driven by multiple motivations, which are monetary and non-monetary incentives related to different subtasks. Based on the crowdsourcing solver's motivation, the incentive mechanisms analyzed and discussed under the two scenarios are targeted to explore the essence of the relationship between the initiator and solver from a new perspective, which is viewed as sustainable crowdsourcing adopted for enterprises.

*3.1. Enterprise Crowdsourcing Single Motivation Incentive Model*

In new product development crowdsourcing process, when monetary incentives are present, solvers rationally evaluate the outcome of their behavior and then adjust their strategies to attain the incentive [47]. According to Liang et al. [48], monetary incentives strengthen the engagement and effort put into tasks. In this section, we assume that the solvers only care about monetary incentives for each subtask in new product development (so-called single motivation). In reality, there are costs incurred in enterprise crowdsourcing activities; in this situation, the initiator will design the crowdsourcing activity incentive mechanisms properly, so that the initiator will maximize its benefit. The incentive mechanism will define how the initiator pays the solver in order to compensate the solver's effort in transferring knowledge in a continuous crowdsourcing activity.

Since the objects of initiator and solver are different, the incentive mechanism should meet two conditions [49,50]. (1) Individual Rationality Constraint (IRC), meaning that the total amount of monetary incentive for the crowdsourcing solver should be no less than solver's Opportunity Cost

(OC), so that solver will continuously keep doing this activity and not turn to others. (2) Incentive Compatibility Constraint (ICC), meaning that the initiator cannot observe the solver's effort level, so the incentive mechanism designed by the initiator should have the solver comply with a rule, stating that the solver will try their best when participating in crowdsourcing in order to meet the expectations of the initiator.

Based on IRC and ICC, this paper establishes a crowdsourcing incentive model in order to achieve maximum gains for the initiator, and keep a sustainable relationship between the initiator and solver throughout the crowdsourcing activity.

Assume the solver's crowdsourcing output gain $\lambda$ is a linear function, expressed in Equation (1):

$$\lambda = \mu e + \delta \tag{1}$$

In the above equation, $e$ is the solver's effort level when participating in an enterprise crowdsourcing knowledge transfer activity. $\mu$ is the marginal output gain when the solver manages to increase one unit of effort level. $\mu$ reflects the quality of the solver's crowdsourcing work and it depends on solver's knowledge transferring ability, so $\mu$ can be regarded as the solver's ability coefficient. $\delta$ is an exogenous random variable of the solver and mainly reflects the random influence from the enterprise operation environment. The mean of $\delta$ is zero and variance of $\delta$ is $\sigma^2$, which means the solver's output is determined by the solver's effort level and the solver's ability coefficient, while it is not impacted by exogenous random variables. $\sigma^2$ is the uncertainty of the exogenous random variables of the solvers, such as unexpected impacts of the enterprise operation environment. The bigger the $\sigma^2$ value, the bigger the impact of the crowdsourcing output due to the higher uncertainty of the enterprise operation or management. The linear function $\lambda$ is the solver's knowledge output and also can be regarded as the initiator's gain.

Assume the initiator makes a linear incentive plan, as expressed in Equation (2):

$$M(\lambda) = \beta\lambda \tag{2}$$

In the above equation, $\beta$ is the incentive coefficient, $0 \leq \beta \leq 1$. $\beta\lambda$ denotes the overall incentive income. If $\beta = 0$, it means the solver does not take any incentive, while $\beta = 1$ means solver takes all incentives in the incentive plan. The incentive coefficient $\beta$ reflects the initiator's incentive strength in relation to the solver. The crowdsourcing solvers can be risk averse, risk neutral or risk loving, and they be can characterized by the value of $\beta$. The initiator is usually the organizer or head of a department in an enterprise, or even the executive of the enterprise, so it is fair to assume that the initiator is risk neutral since the initiator has the ability to take some risks during crowdsourcing to achieve the maximum amount of profit. According to expected utility theory [51], risk neutral means that the expected utility is equal to the expected income. So, the initiator's expected utility ($E_i$) is equal to initiator's income, as shown in the below equation:

$$E_{in} = E(\lambda - M(\lambda)) = E(\lambda - \beta\lambda) \tag{3}$$

By combining Equations (1) and (3), the initiator's expected utility can be expressed as follows:

$$E_{in} = (1 - \beta)\mu e \tag{4}$$

When participating in a crowdsourcing activity, the solver will spend time and share knowledge, and this can be viewed as an Effort Cost, and the solver's Effort Cost can be expressed as follows:

$$C_E = \frac{te^2}{2} \tag{5}$$

in Equation (5), *t* is the Effort Cost coefficient and the value of *t* is larger than zero. For the same Effort Cost with a value of *e*, a larger *t* indicates that the solver expends more effort, which is a negative utility.

Besides the Effort Cost (EC) $C_E$, since solvers are diversified in terms of risk aversion, it is necessary to add risk cost into the overall cost of the solver. The risk cost $C_R$ can be expressed as follows:

$$C_R = \frac{\varepsilon\beta^2\sigma^2}{2} \tag{6}$$

where $\varepsilon$ is the degree of risk aversion for the solver, a larger $\varepsilon$ means that the solver tends to be more risk averse.

With Equations (2), (5) and (6), the solver's certainty equivalence income can be expressed as follows:

$$E_s = E(M(\lambda)) - C_E - C_R = \beta\mu e - \frac{te^2}{2} - \frac{\varepsilon\beta^2\sigma^2}{2} \tag{7}$$

Assume the solver's Opportunity Cost for participating in enterprise crowdsourcing is $C_o$; according to IRC, the solver's certainty equivalence income $E_s$ should be no less than the Opportunity Cost $C_o$, which means:

$$IRC : \beta\mu e - \frac{te^2}{2} - \frac{\varepsilon\beta^2\sigma^2}{2} \geq C_o \tag{8}$$

On the other hand, for ICC, based on the incentive plan provided by initiator, the solver always determines the effort level *e* to maximize the expected utility function. Therefore, the initiator expects that the solver can participate with effort level *e* and the expected utility the solver can obtain from this crowdsourcing activity should be no less than the expected utility the solver can obtain from other activities. The ICC can be expressed as follows:

$$ICC : e \in \max(\beta\mu e - \frac{te^2}{2}) \tag{9}$$

Based on the above assumptions, in order to provide a clear and thorough understanding of the model details, it is necessary to summarize all the parameters and descriptions together, as shown in Table 3:

**Table 3.** Notations for single motivation incentive model.

| Parameter | Description |
|---|---|
| $\beta$ | Solver's incentive coefficient |
| $\lambda$ | Solver's total output gain |
| $\mu$ | Solver's marginal output gain, reflecting solver's work quality |
| $e$ | Solver's effort level |
| $\delta$ | Exogenous random variables to solver, reflecting enterprise operation environment |
| $\sigma^2$ | Uncertainty of exogenous random variables to solver |
| $\varepsilon$ | Solver's degree of risk aversion |
| $t$ | Solver's Effort Cost coefficient |
| $M(\lambda)$ | Initiator's payment to solver |
| $E_{in}$ | Initiator's expected income |
| $E_s$ | Solver's certainty equivalence income |
| $C_E$ | Solver's Effort Cost |
| $C_R$ | Solver's risk cost |
| $C_o$ | Solver's Opportunity Cost |

When the initiator and solver establish the Principal–Agent relationship, the initiator is the principal and the solver is the agent. Enterprise crowdsourcing cooperation is like a contractual relationship, which means that the solver will provide a product or service, while the initiator should pay for it. As mentioned earlier, the initiator is facing a large group of unknown and heterogeneous

people or organizations (the "crowd"), which means that the crowd's efforts are not observed by initiator, leading to an asymmetrical information situation.

The asymmetrical information condition can be transformed into a mathematic problem, as shown below:

$$\begin{cases} MaxE_{in} = (1-\beta)\mu e \\ s.t. \\ IRC: \beta\mu e - \frac{te^2}{2} - \frac{\varepsilon\beta^2\sigma^2}{2} \geq C_o \\ ICC: e \in \max(\beta\mu e - \frac{te^2}{2}) \\ \mu, e, t, \varepsilon, \sigma \geq 0 \\ 0 \leq \beta \leq 1 \end{cases} \tag{10}$$

The solution process for Equation (10) is shown in Appendix A. The best incentive coefficient $\beta$* and solver's effort level $e$* can be derived as follows:

$$\begin{cases} \beta^* = \frac{1}{1+t\varepsilon\sigma^2\mu^{-2}} \\ e^* = \frac{u}{t(1+t\varepsilon\sigma^2\mu^{-2})} \end{cases} \tag{11}$$

As for the incentive coefficient $\beta$* in Equation (11), it is interesting to see how each factor has an impact, such as the solver's work quality, their Effort Cost, the solver's risk preference, the enterprise operation environment, etc. Through the deviation of the incentive coefficient $\beta$*, the following result can be obtained:

$$\begin{cases} \frac{\partial\beta^*}{\partial t} = -\frac{\varepsilon\sigma^2\mu^{-2}}{(1+t\varepsilon\sigma^2\mu^{-2})^2} \leq 0 \\ \frac{\partial\beta^*}{\partial\varepsilon} = -\frac{t\sigma^2\mu^{-2}}{(1+t\varepsilon\sigma^2\mu^{-2})^2} \leq 0 \\ \frac{\partial\beta^*}{\partial\sigma^2} = -\frac{t\varepsilon\mu^{-2}}{(1+t\varepsilon\sigma^2\mu^{-2})^2} \leq 0 \\ \frac{\partial\beta^*}{\partial\mu} = \frac{2t\varepsilon\sigma^2\mu^{-3}}{(1+t\varepsilon\sigma^2\mu^{-2})^2} \geq 0 \end{cases} \tag{12}$$

Equation (12) indicates that the solver's incentive coefficient provided by the initiator has different impacts on those factors, which can be described as follows:

**Proposition 1.** *For the crowdsourcing solver, the incentive coefficient is positively correlated to the solver's marginal output, while the incentive coefficient is negatively impacted by the solver's Effort Cost, degree of risk aversion and the uncertainty of exogenous random variables in the enterprise.*

*3.2. Enterprise Crowdsourcing Multiple Motivations Incentive Model*

In crowdsourcing, monetary incentives are important for attracting solvers' participation, while non-monetary incentives, such as learning new knowledge, entertainment, building networks, etc., are also critical factors to motivate solvers. Actually, because of these non-monetary factors, traditional crowdsourcing tends to evolve into a sustainable crowdsourcing mode, such as in the case of Threadless, which designs new products such as T-shirts for an online community crowd. According to Tran et al. [11], the solvers are driven either by financial rewards or non-monetary factors (or even both) for different tasks. Some solvers become the "Prosumer", a combination of "pro" and "consumer", which means they want to join the co-creation process to provide input and have influence in product design. This is an interesting trend for new product development and commercialization, and such an open innovation model contributes to a healthy sustainable crowdsourcing process.

Solvers are driven by multiple incentives for different subtasks within sustainable crowdsourcing, which means they are not only driven by monetary extrinsic incentives, but also by non-monetary intrinsic incentives, such as learning new knowledge, gaining experiences, self-esteem, etc. We call this solvers' multiple motivations. In order to make the model simple, the multiple motivations are defined

as money incentives and non-money incentives. As in Section 3.1, it is assumed that the crowdsourcing initiator is risk neutral and solvers can be risk averse, risk neutral or risk loving.

The solver's effort level $e$ is defined as a vector, $e = (e_1, e_2)$, $e_1$ is the effort level for a monetary incentive and $e_2$ is effort level for a non-monetary incentive. For simplicity, define $e = \begin{bmatrix} e_1 \\ e_2 \end{bmatrix}$, and assume there is an observable information vector $Z$, which is a function of effort level $e$. $Z$ is defined as below:

$$Z(e_1, e_2) = e + \delta \tag{13}$$

$\delta$ is an exogenous random variable which follows normal distribution $N(0, \sigma^2)$. $Z = (z_1, z_2)$ can be expressed as:

$$z_i = e_i + \delta_i (i = 1, 2) \tag{14}$$

As with the single motivation incentive model, it is assumed that the initiator provides a linear incentive plan and payment $M(e_1, e_2)$ with incentive coefficient $\beta = (\beta_1, \beta_2)^T$, which is:

$$M(e_1, e_2) = \beta^T Z \tag{15}$$

The expected payment to the solver is:

$$E(M(e_1, e_2)) = \beta^T e \tag{16}$$

Assume that the solver's effort output gain is $G(e_1, e_2)$ and their Effort Cost is $C_E(e_1, e_2)$. The initiator's expected income $E_{in}$ and solver's certainty equivalence income $E_S$ can be expressed as follows:

$$E_{in} = G(e_1, e_2) - \beta^T e \tag{17}$$

$$E_s = \beta^T e - C_E(e_1, e_2) - \frac{\varepsilon \beta^T \sigma^2 \beta}{2} \tag{18}$$

In Equation (18), $\frac{\varepsilon \beta^T \sigma_i^2 \beta}{2}$ is the solver's risk cost, and $\varepsilon$ is the degree of risk aversion for the solver; a larger $\varepsilon$ value means that the solver tends to be more risk averse.

The crowdsourcing cooperation should meet the IRC condition, which is:

$$IRC : \beta^T e - C_E(e) - \frac{\varepsilon \beta^T \sigma^2 \beta}{2} \geq C_o \tag{19}$$

For ICC, it is:

$$ICC : (e_1, e_2) \in \text{argmax} \left\{ \beta^T e - C_E(e_1, e_2) \right\} \tag{20}$$

Based on the above assumptions, the notations for the multiple motivation incentive model are listed in Table 4.

**Table 4.** Notations for single motivation incentive model.

| Parameter | Description |
|---|---|
| $\beta$ | Solver's incentive coefficient vector provided by initiator, including extrinsic ($\beta_1$) and intrinsic ($\beta_2$). |
| $G(e_1, e_2)$ | Solver's output by two efforts |
| $e(e_1, e_2)$ | Solver's effort level |
| $\delta$ | Exogenous random variables to solver, reflecting enterprise operation environment |
| $\sigma^2$ | Uncertainty of exogenous random variables in enterprise to solver |
| $\varepsilon$ | Solver's degree of risk aversion |
| $Z$ | Observable information vector for solver's effort |
| $M(e_1, e_2)$ | Initiator's payment to solver |
| $E_{in}(e_1, e_2)$ | Initiator's expected income |
| $E_s(e_1, e_2)$ | Solver's certainty equivalence income |
| $C_E(e_1, e_2)$ | Solver's Effort Cost |
| $C_o$ | Solver's Opportunity Cost |

Based on the nature of enterprise crowdsourcing, the initiator cannot observe the solver's effort level for both monetary incentives and non-monetary incentives. In this case, both the IRC and ICC rules should be satisfied. The initiator will provide the incentive coefficient vector to maximize the initiator's expected income $E_i$. This can be transformed into a mathematical problem, as shown below:

$$\begin{cases} MaxE_{in} = G(e_1, e_2) - \beta^T e \\ s.t. \\ IRC : \beta^T e - C_E(e_1, e_2) - \frac{\varepsilon \beta^T \sigma^2 \beta}{2} \geq C_o \\ ICC : (e_1, e_2) \in \text{argmax}\{\beta^T e - C_E(e_1, e_2)\} \\ e_i, \varepsilon, \sigma_i \geq 0 \\ 0 \leq \beta_i \leq 1 \end{cases} \tag{21}$$

For the solver, the best incentive coefficient vector $\beta^*$ is shown below. The solution process is shown in Appendix B.

$$\beta^* = (I + \varepsilon[C_{Eij}]\sigma^2)^{-1} G' \tag{22}$$

where $I$ is the unit matrix, $G' = (G_1', G_2')^T$ is a first order partial derivative vector. $G_i' = \frac{\partial G_i}{\partial e_i}$ is the marginal output gain w.r.t. the $i$th's work effort, and $[C_{Eij}] = \frac{\partial \beta}{\partial e^T}$ is the rate of incentive change per the solver's unit effort.

The result of Equation (22) can be described as follows:

**Proposition 2.** *Considering the crowdsourcing solver's multiple motivations, the initiator can find the best incentive coefficient vector to maximize the profit. The incentive coefficient vector is $\beta^* = (I + \varepsilon[C_{Eij}]\sigma^2)^{-1} G'$.*

From the initiator's point of view, it is interesting to examine the difference of monetary incentive coefficient $\beta_1$ and non-monetary incentive coefficient $\beta_2$ since, in the end, the initiator needs decide the management approach to keep the solver's momentum and eventually achieve a sustainable crowdsourcing process.

Since the uncertainty of exogenous random variables for monetary incentives can be observed by the initiator, who can monitor and calculate the crowdsourcing spending and budget, the uncertainty of exogenous random variables $\sigma_1^2$ in $z_1$ is zero. The uncertainty of exogenous random variables of non-monetary incentive information $\sigma_2^2$ is due to the solver's private information, which the initiator cannot observe. So, Equation (22) can be transformed as below:

$$\begin{bmatrix} \beta_1 \\ \beta_2 \end{bmatrix} = \left[ \begin{bmatrix} 1 & 0 \\ 0 & 1 \end{bmatrix} + \varepsilon \begin{bmatrix} C_{E11} & C_{E12} \\ C_{E21} & C_{E22} \end{bmatrix} \begin{bmatrix} 0 & 0 \\ 0 & \sigma_2^2 \end{bmatrix} \right]^{-1} \begin{bmatrix} G_1' \\ G_2' \end{bmatrix} \tag{23}$$

The solution to Equation (23) is as follows:

$$\begin{cases} \beta_1 = G_1' - G_2' \frac{\varepsilon C_{E12} \sigma_2^2}{1 + \varepsilon C_{E22} \sigma_2^2} \\ \beta_2 = \frac{G_2'}{1 + \varepsilon C_{E22} \sigma_2^2} \end{cases} \tag{24}$$

Since $\beta_1 \geq 0$, so $G_1' \geq G_2' \frac{\varepsilon C_{E12} \sigma_2^2}{1 + \varepsilon C_{E22} \sigma_2^2}$, which means that when the solver's monetary marginal output gain $G_1'$ is no less than a critical condition ($G_2' \frac{\varepsilon C_{E12} \sigma_2^2}{1 + \varepsilon C_{E22} \sigma_2^2}$), the initiator will provide a positive monetary incentive. When $G_1$ is less than the critical condition, the $\beta 1$ will be less than zero, so the initiator will not provide any monetary incentive. At the same time, both monetary and non-monetary incentive coefficients are negatively correlated to the non-monetary uncertainty of exogenous random variables $\sigma_2^2$ in the enterprise.

## 4. Simulation and Discussion

In order to provide a direct and straightforward explanation of the above formulations and analyses, in this section, some numerical simulation examples, with presumptions of some of the parameters' values, are presented to confirm the above results.

### 4.1. Solver's Work Quality Impact to Initiator Profit and Incentive Coefficient

For enterprise crowdsourcing activity, high quality outputs from solvers are rare and are critical for sustainable ongoing crowdsourcing over time [5]. Based on the model in this paper, it is necessary to analyze and simulate how the solver's work quality impacts the incentive coefficient and initiator profit.

This session uses the monetary single motivation as example. Based on result of Equations (10) and (11), it is assumed that the value of the parameters in the equations are set in Table 5, as shown below.

**Table 5.** Parameter presumptions for solver's work quality simulation.

| Parameter | Description | Value |
|-----------|-------------|-------|
| $\sigma^2$ | Uncertainty of exogenous random variables to solver | 0.5 |
| $\varepsilon$ | Solver's degree of risk aversion | 0.3 |
| $t$ | Solver's Effort Cost coefficient | 0.8 |
| $C_o$ | Solver's Opportunity Cost | 10 |

With the above parameters' presumed values, the simulation results for the solver's marginal output gain, which reflects the solver's work quality and incentive coefficient, as well as the initiator's profits, are shown as below:

From the simulation results, it is clear to see that the solver's work quality is positively correlated to both the incentive coefficient and the initiator's profit. Naturally, a higher quality output will lead to a more sustainable crowdsourcing activity, so the initiator will gain more profit from a longer and higher quality crowdsourcing process, which will eventually bring a higher return, as the incentive coefficient in the paper, to the solver as well. This conclusion is similar to Liao's findings based on the data collected from Topcoder.com. Liao et al. [52] found that the solver's work effort has a positive impact on the amount of compensation in crowdsourcing. However, the results in this paper are interpreted and applied in a better way in terms of the quantified model results.

It is also interesting to observe, from Figure 1a, that the initiator will have higher profit when solver's effort is higher. In particular, when the solver's work quality is lower at some points, the profit will be negative. We define this point as Quality Equilibrium Point (QEP), which indicates the criteria for the initiator to judge and choose from proper solvers during sustainable crowdsourcing. In Figure 1b, although the solver's work quality significant impacts the incentive coefficient at the beginning stage, it will quickly reach the maximum value, which is one. We define this turning point as the Quality Saturation Point (QSP), which is also important to the initiator when designing the incentive plan. This is a tradeoff situation between the initiator and solver. On one hand, from the crowdsourcing process point of view, initiator should pick the solver with the highest work quality to improve the crowdsourcing cooperation profit; on the other hand, solvers will only tend to improve their work quality to the QSP in order to obtain the maximum incentive coefficient.

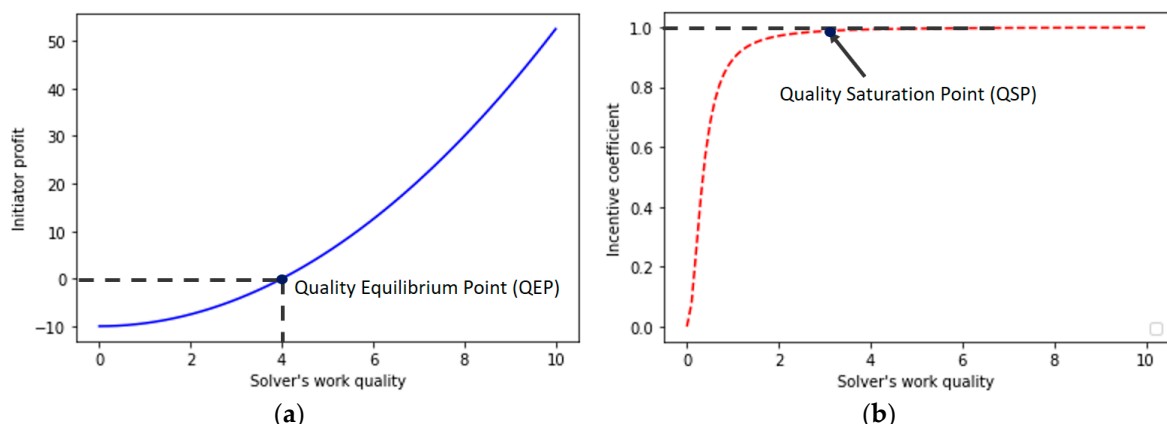

**Figure 1.** Simulation result for single motivation incentive situation. (**a**) Initiator profit vs. solver's work quality; (**b**) incentive coefficient vs. solver's work quality.

### 4.2. How Does Uncertainty of Enterprise Operation Environment Affect the Incentive Coefficient?

Crowdsourcing, as an innovative operation management approach to enterprise, attracts many individuals to participate in activities or contests. Organizational and environmental factors are important to the individual's success [53], which also impacts on the profit obtained from this innovative process.

For the asymmetrical information situation, as indicated in Equations (11) and (22), the uncertainty of exogenous random variables matters. From the crowdsourcing solver's perspective, the uncertainty of exogenous random variables is mainly due to the enterprise operation environment, which is one of the important factors that can impact crowdsourcing success [54,55]. In order to provide a clear assessment of the effect of enterprise operation environment, we assume the value of some parameters in the model are set as shown in Table 6:

**Table 6.** Parameter presumptions for uncertainty of enterprise operation environment.

| Parameter | Description | Value |
|:---:|:---|:---:|
| $\mu$ | Solver's marginal output gain, reflecting solver's work quality | 2 |
| $\varepsilon$ | Solver's degree of risk aversion | 0.3 |
| $t$ | Solver's Effort Cost coefficient | 0.8 |
| $G_1$ | Marginal output gain by monetary incentive | 0.8 |
| $G_2$ | Marginal output gain by non-monetary incentive | 0.4 |
| $C_{E12}$ | Monetary incentive change per solver's non-monetary unit effort | 0.5 |
| $C_{E22}$ | Non-monetary incentive change per solver's non-monetary unit effort | 0.5 |

Based on the Equations (11) and (22) and the presumed parameters in Table 6, the simulation results are shown as below in Figure 2:

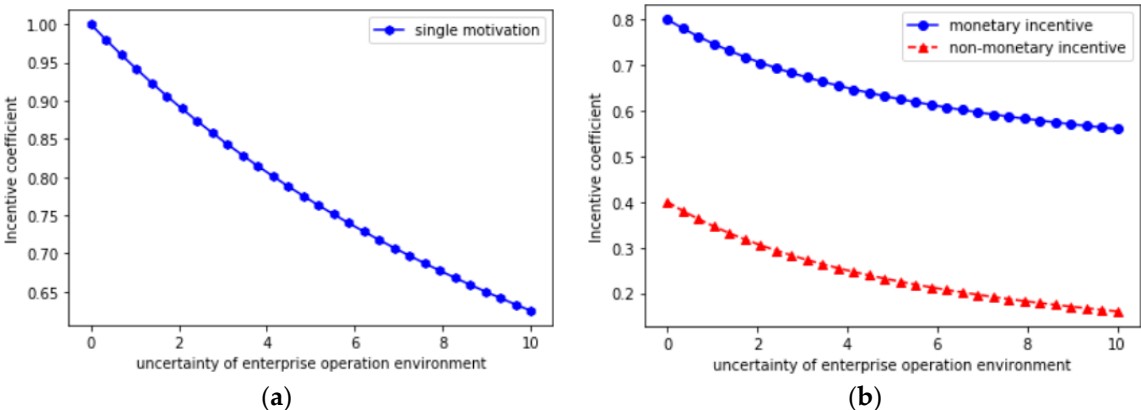

**Figure 2.** Incentive coefficient vs. uncertainty of enterprise operation environment. (**a**) single motivation incentive; (**b**) multiple motivations incentive.

Uncertainty can be defined as unpredictable events that disturb operations and the performance of an enterprise [56], and it could result in the tardy delivery of enterprise products [57]. The environmental uncertainty may also influence organizational requirements with respect to information processing [58]. The impacts of an uncertain environment on crowdsourcing have not yet been analyzed. From the simulation, it is clear to see that the solver's crowdsourcing incentive coefficients are negatively impacted by the uncertainty of the external operation environment. In order to build sustainable crowdsourcing in enterprise, the management team should maintain smooth operation and provide information in a way that is as transparent as possible to participants.

*4.3. The Monetary Incentive Plan When Considering Solver's Effort Cost*

Enterprise crowdsourcing can contribute to solving technical problems, creating innovations and optimizing the cost of an organization's activity. Despite the promising output, the solver's Effort Cost is not neglected, especially in sustainable crowdsourcing. Considering Equations (11) and (24), it is assumed the value of the parameters in the equations are set as shown in Table 7.

**Table 7.** Parameter presumptions for solver's Effort Cost.

| Parameter | Description | Value |
|---|---|---|
| $\mu$ | Solver's marginal output gain, reflecting solver's work quality | 2 |
| $\varepsilon$ | Solver's degree of risk aversion | 0.3 |
| $\sigma^2$ | Uncertainty of exogenous random variables in enterprise | 0.5 |
| $C_o$ | Solver's Opportunity Cost | 10 |
| $G_1$ | Marginal output gain by monetary incentive | 0.8 |
| $G_2$ | Marginal output gain by non-monetary incentive | 0.4 |

With the above presumed parameters value, the simulations between the incentive coefficient and solver's Effort Cost are shown in Figure 3.

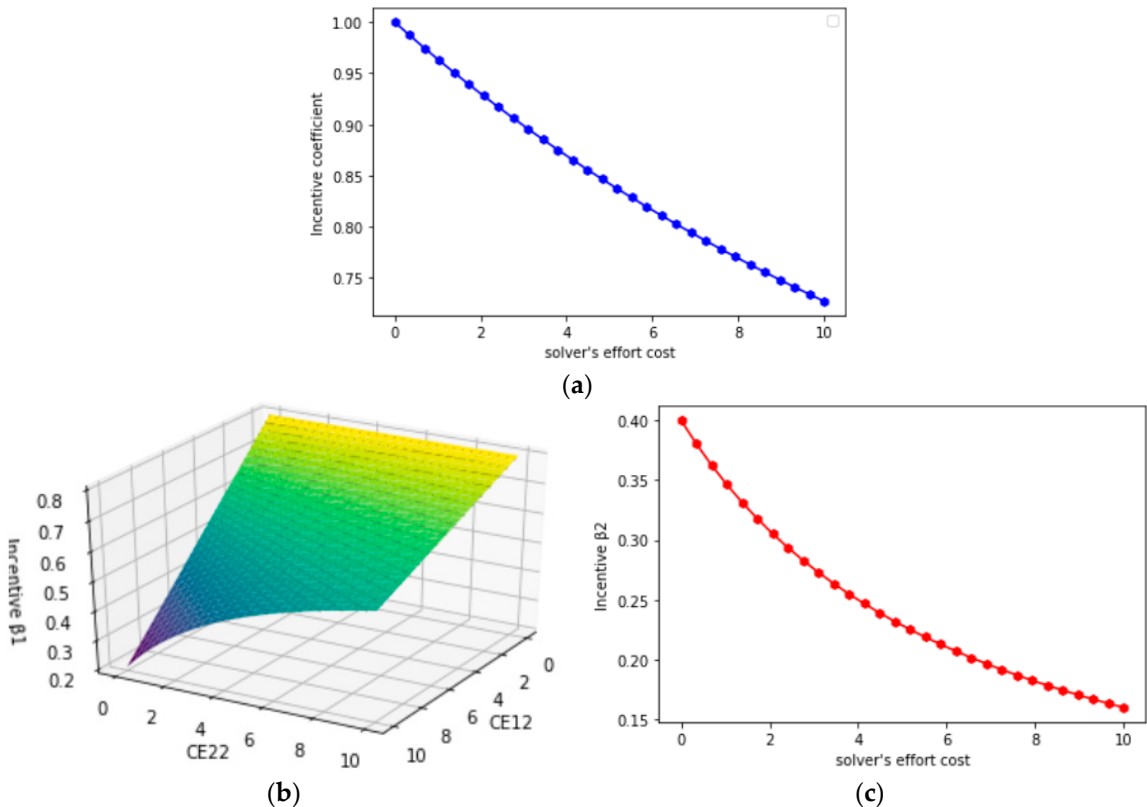

**Figure 3.** Monetary incentive coefficient vs. solver's Effort Cost. (**a**) single motivation incentive; (**b**) multiple motivations incentive $\beta 1$; (**c**) multiple motivations incentive $\beta 2$.

For the single motivation case, as can be seen from Figure 3a, the solver's Effort Cost has a clear negative impact on the incentive coefficient—the higher the solver's cost, the smaller the incentive coefficient. This result also indicates that, in order to have a better incentive effect, the initiator tends to provide higher incentive coefficients to lower cost solvers, such as junior participants. This finding aligns with the previous research result [21] that 90% of the solutions a company received through crowdsourcing challenges were from junior to mid-level members. In addition, a similar conclusion can also be obtained in the crowdsourcing supply chain [59], showing that the whole organization's profit is negatively correlated to the solver's proportional profit.

It is interesting to look at the results from multiple motivations incentive in Figure 3b,c. The non-monetary incentive coefficient $\beta 2$ is negatively correlated to the rate of non-monetary incentive change per the solver's unit effort driven by non-monetary incentives. The initiator tends to provide lower non-monetary incentives to solvers with a higher non-monetary Effort Cost, which sometimes means a slower learning speed, or more difficult to share knowledge. The monetary incentive coefficient $\beta 1$ is negatively correlated to $C_{E12}$ and positive to $C_{E22}$. $C_{E12}$ means the rate of monetary incentive change per solver's unit effort driven by non-monetary incentives, while $C_{E22}$ means the rate of non-monetary incentive change per the solver's unit effort driven by non-monetary incentives. When the solver is driven by both intrinsic and extrinsic motivations, if the initiator is more sensitive to the benefits of the solver's intrinsic efforts, the solver's monetary incentive will be higher. This conclusion will guide the solver to focus more on intrinsic motivations [60], such as learning new knowledge, sharing experiences, etc.

## 5. Conclusions

Sustainable crowdsourcing is a promising approach to acquire distant knowledge and improve an organization's innovation ability. While many enterprises pilot crowdsourcing for technical problems,

only a few have embedded it today as a routine sourcing strategy in their innovation activities [61]. Prior studies [18,36,43] concluded that the solver's motivation is a hot topic for crowdsourcing and is critical to achieving sustainable crowdsourcing. However, there are few papers that provide guidance on how to set proper incentive mechanisms to satisfy the solver's motivation in a quantitative way. This research stems from practical application scenarios in enterprise crowdsourcing, especially when incentive mechanisms play a key role in driving the activity in order to sustainably achieve a result for both the initiator and solver. This paper discusses the sustainable crowdsourcing solver's incentive mechanisms based on PAM, under the conditions of a single motivation and multiple motivations. Through comparative analysis and numerical simulations, the following conclusions can be drawn:

(1) The solver's marginal output gain, which can be interpreted as the solver's work quality, plays a key role in impacting enterprise profit and the incentive coefficient. On the one hand, the higher the work quality from the solver, the more benefits the initiator gets. This conclusion is common sense. On the other hand, the solver's work quality will also influence the value of the incentive coefficient provided by the initiator. For rational participants, the solvers will determine the work effort and quality when they observe the initiator's incentive plan, while the initiator will also estimate the expected profit from the crowdsourcing solver's work effort, then decide the incentive for the solvers. The new findings in this paper provide managerial references for enterprise. From a sustainable crowdsourcing point of view, we recommend that the initiator can probably define a key performance index (KPI) to quantify and monitor the solver's work quality, so that the QEP and QSP points are calculated, and enterprise can predict the expected profit and provide a proper incentive plan for the solver based on the results in this paper;

(2) The incentive coefficient will be partially impacted by the external uncertainties that solvers are facing during enterprise sustainable crowdsourcing. The solvers will be expected to obtain less incentives when there are more uncertainties, such as uncertainties from enterprise research and development process. This conclusion is a good indication from an enterprise management point of view. In order to maintain a healthy sustainable crowdsourcing process, though the enterprise cannot control the external industry environment, we urge enterprise managers to try to take efforts to reduce the uncertainties of enterprise product research and development processes, such as identifying the purpose of the research, aligning internal sponsors and external material suppliers, stating product specifications, organizing regular meeting to review the schedule and discuss technical issues, etc. In fact, these actions not only benefit sustainable crowdsourcing, but also can strengthen the enterprise's long term competitive edge;

(3) When participating in different subtasks in sustainable crowdsourcing, no matter whether the solver is driven by single motivation or multiple motivations, the solver's Effort Cost has a negative impact on the monetary incentive coefficient. The solver's Effort Cost will influence the incentive plan, and the initiator tends to provide higher monetary incentive offerings to lower Effort Cost solvers. Furthermore, the solver is expected to obtain a higher incentive if the solver has a higher non-monetary incentive changing rate per the solver's unit effort, which usually means the solver is a quicker learner, or an effective knowledge producer. This conclusion is especially useful for sustainable crowdsourcing in internal processes [31], in which the target crowd will be focused on junior and mid-level employees. Along this line of consideration, in order to reduce the solver's Effort Cost, we recommend that the initiator to creates a good knowledge learning and sharing atmosphere when carrying out sustainable crowdsourcing processes in enterprise, which will inspire the solvers with multiple motivations to achieve great outputs. From the solver's point of view, more attention should be paid to non-monetary output, such as gaining new knowledge, developing technical skills, etc. Therefore, the initiator should build a virtuous circle of solvers in enterprise sustainable crowdsourcing.

The conclusions obtained by using the Principal–Agent Model in this research present important practical guidance for enterprise to execute sustainable crowdsourcing within organizations.

The methods and results in this paper not only enrich the research of enterprise crowdsourcing incentive mechanisms [62–64], but also extend the application of PAM [65–67]. In addition, the quantitative analysis is a supplementary approach for prior qualitative crowdsourcing incentive mechanism research [68,69]. From a model and approach perspective, the PAM used in the research is well-adapted for sustainable crowdsourcing processes. The relationship between the initiator and solver is similar to the one between the principal and agent. The other advantage of using PAM in the research is that the model can precisely simulate the solver's incentive with different factors that have practical management implications. However, the model is not ideal and one of the drawbacks is that, in order to use the model to guide actual operations in reality, it requires that the enterprise has a strong operational capability to quantify some indexes, like the solver's work effort, Effort Cost, etc.

Despite the richness of the conclusions and approach, it should be noted that the present research is by no means free of limitations. For example, we assume the incentive plan is a linear function, while, in reality, besides the linear function plan, the enterprise may provide other types of incentive plans, such as a base salary with additional bonuses, or a discrete function with different levels, etc. Furthermore, when a product designed by sustainable crowdsourcing is in the commercial phase, it will be interesting to explore an integrated incentive model, considering the product price and sales revenue. Future research should examine these patterns in greater detail.

**Author Contributions:** The main activities of the team of authors can be described as follows. L.Y. and G.W. designed the didactic scenario, initiated the study and were significantly involved in its conception. G.W. wrote the draft of the article, designed and refined the study from a didactic and motivational perspective. Conceptualization, L.Y.; investigation, G.W.; resources, G.W. and L.Y.; writing—original draft preparation, G.W.; writing—review and editing, L.Y.; supervision, L.Y.; project administration, L.Y.; funding acquisition, L.Y. All authors have read and agreed to the published version of the manuscript.

**Funding:** The works that are described in this paper are funded by the General Program of National Natural Science Foundation of China, "Research on Random Symmetrical Cone Complementarity Problems and Related Topics" (No: 11671250).

**Acknowledgments:** The authors thank the reviewers for their careful reading and providing some pertinent suggestions.

**Conflicts of Interest:** The authors declare no conflicts of interest. The funders had no role in the design of the study; in the collection, analyses, or interpretation of data; in the writing of the manuscript, or in the decision to publish the results.

## Abbreviations

| | |
|---|---|
| PAM | Principal–Agent Model |
| IRC | Individual Rationality Constraint |
| ICC | Incentive Compatibility Constraint |
| EC | Effort Cost |
| OC | Opportunity Cost |
| QEP | Quality Equilibrium Point |
| QSP | Quality Saturation Point |
| KPI | Key Performance Index |

## Appendix A

As mentioned in Section 3.1, we analyzed the best incentive coefficient under the asymmetrical information situation, with solvers having a single motivation. Below are the steps to the solution.

Step 1: List the optimization function and subject of the condition as follows:

$$\begin{cases} MaxE_{in} = (1-\beta)\mu e \\ s.t. \\ IRC : \beta\mu e - \frac{te^2}{2} - \frac{\varepsilon\beta^2\sigma^2}{2} \ge C_o \\ ICC : e \in \max(\beta\mu e - \frac{te^2}{2}) \\ \mu, e, t, \varepsilon, \sigma \ge 0 \\ 0 \le \beta \le 1 \end{cases}$$

Step 2: The asymmetry information means the crowdsourcing solver's effort level $e$ cannot be observed; according to the ICC rule, the solver would get the maximum certainty equivalence income effort $E_S$ when the solver spent the effort level $e$. So, by derivation of above equation, the ICC can be expressed as follows:

$$e = \frac{\mu\beta}{t}$$

Step 3: The IRC inequation can be transformed as below:

$$\beta\mu e \ge C_o + \frac{te^2}{2} + \frac{\varepsilon\beta^2\sigma^2}{2}$$

Step 4: For the inequation of step 3, put the maximum value of $\beta\mu e$ into $E_{in}$, and get the updated optimization function:

$$MaxE_{in} = \mu e - (C_o + \frac{te^2}{2} + \frac{\varepsilon\beta^2\sigma^2}{2})$$

while considering the result $e = \frac{\mu\beta}{t}$ in Step 2, the $E_{in}$ can be transformed as below:

$$MaxE_{in} = \frac{\mu^2\beta}{t} - C_o - \frac{\mu^2\beta^2}{2t} - \frac{\varepsilon\beta^2\sigma^2}{2}$$

Step 5: Derive the optimization function from $\beta$ and set the formula as 0; the result can be derived as follows:

$$\begin{cases} \beta^* = \frac{1}{1+t\varepsilon\sigma^2\mu^{-2}} \\ e^* = \frac{u}{t(1+t\varepsilon\sigma^2\mu^{-2})} \end{cases}$$

Quod Erat Demonstrandum (Q.E.D.).

**Appendix B**

As mentioned in Section 3.2, we analyzed the best incentive coefficient under the asymmetrical information situation, with solvers having multiple motivations, including monetary motivations and non-monetary motivations. Below are the steps to the solution.

Step 1: List the optimization function and subject of the condition as follows:

$$\begin{cases} MaxE_{in} = G(e_1, e_2) - \beta^T e \\ s.t. \\ IRC : \beta^T e - C_E(e_1, e_2) - \frac{\varepsilon\beta^T\sigma^2\beta}{2} \ge C_o \\ ICC : (e_1, e_2) \in \text{argmax}\{\beta^T e - C_E(e_1, e_2)\} \\ e_i, \varepsilon, \sigma_i \ge 0 \\ 0 \le \beta_i \le 1 \end{cases}$$

Step 2: Since $e_i$ is positive and larger than zero, derive the ICC from $e_i$ and set the formula as 0. As illustrated in Section 3.2, for simplicity, we define $e = \begin{bmatrix} e_1 \\ e_2 \end{bmatrix}$, so the equation can be expressed as

$$\beta = \frac{\partial C_E(e)}{\partial e^T}$$

Define $[C_{Eij}] = \frac{\partial \beta}{\partial e^T}$, which can be expanded as the matrix below:

$$[C_{Eij}] = \begin{bmatrix} \frac{\partial \beta_1}{\partial e_1} & \frac{\partial \beta_1}{\partial e_2} \\ \frac{\partial \beta_2}{\partial e_1} & \frac{\partial \beta_2}{\partial e_2} \end{bmatrix} = \begin{bmatrix} C_{E11} & C_{E12} \\ C_{E21} & C_{E22} \end{bmatrix}$$

Step 3: The IRC inequation can be transformed as:

$$\beta^T e \geq C_E(e_1, e_2) + \frac{\varepsilon \beta^T \sigma^2 \beta}{2} + C_o$$

Step 4: Put the maximum value of $\beta^T e$ into $E_{in}$, and get the updated optimization function:

$$MaxE_{in} = G(e_1, e_2) - C_E(e_1, e_2) - \frac{\varepsilon \beta^T \sigma^2 \beta}{2} - C_o$$

Step 5: Derive the optimization function from $\beta$ and set the formula as 0:

$$\frac{\partial G(e)}{\partial \beta} - \frac{\partial C_E(e)}{\partial \beta} - \frac{1}{2} \frac{\partial \varepsilon \beta^T \sigma^2 \beta}{\partial \beta} = 0$$

The above equation can be transformed as below:

$$\frac{\partial G(e)}{\partial e^T} \frac{\partial e^T}{\partial \beta} - \frac{\partial C_E(e)}{\partial e^T} \frac{\partial e^T}{\partial \beta} - \frac{1}{2} \frac{\partial \varepsilon \beta^T \sigma^2 \beta}{\partial e^T} \frac{\partial e^T}{\partial \beta} = 0$$

while $\beta = \frac{\partial C_E(e)}{\partial e^T}$, set $G' = \frac{\partial G}{\partial e^T}$, $[C_{Eij}] = \frac{\partial \beta}{\partial e^T}$, then the solution can be shown as below:

$$\beta^* = (I + \varepsilon [C_{Eij}] \sigma^2)^{-1} G'$$

Quod Erat Demonstrandum (Q.E.D.).

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
