# Peer review of "Analysis of Enterprise Sustainable Crowdsourcing Incentive Mechanism Based on Principal-Agent Model"

_sustainability, doi:10.3390/su12083238_

Round 1
Reviewer 1 Report
The topic of the research is interesting and very important for business, scientists, government bodies and consumers/users. The article provides valuable information and has great potential, but in its current state it lacks precision and refinement.
My reservations apply:
1) The approach to the literature review is mainly journalistic and the information provided is general.
(2) The model presented has no value in terms of management and application possibilities in enterprises. It has a mathematical nature, for which a model that can be easily applied in business should be built.
3) Discussion with the results of existing studies - there is currently no reference to the results of other authors' studies referring to the authors' results.
Author Response
Dear Professor:
First of all, thank you very much for your feedback! These are very good suggestions and we fully agree with your points. Here are our responses in terms of your feedback:
(In order to align the line No., suggest you can use “Tracking Changes = All Markup”)
The topic of the research is interesting and very important for business, scientists, government bodies and consumers/users. The article provides valuable information and has great potential, but in its current state it lacks precision and refinement.
My reservations apply:
1) The approach to the literature review is mainly journalistic and the information provided is general.
==> We reviewed the literature again and understand your feedback are absolutely correct. Now we updated the literature by moving some paragraphs (Line#95-102, and line#208-213) to “Introduction” and “Model Formulation” parts, adding one paragraph and more references (line#194-207) to elaborate current approaches, adjusting the crowdsourcing literature review content to line #103-118. We even add one table (Table 2.) to list current key motivations in literature review. So that the literature review part is more structural to include crowdsourcing, detail motivation and specific research approaches.
2) The model presented has no value in terms of management and application possibilities in enterprises. It has a mathematical nature, for which a model that can be easily applied in business should be built.
==> Thank you for your feedback, very appreciate! In fact, this research stems from the enterprise crowdsourcing application in my real work. One of the authors has been working as a management role in a high-tech (semiconductor industry) company for years and there were many opportunities to employ crowdsourcing in new product development projects. In those projects, the organization and participants are contractual relationship and the incentive mechanism is indeed critical to achieve the better performance for sustainable crowdsourcing. The Principal-Agent Model we used is a classical contractual relationship model, which is suitable for studying the design of incentive mechanism based on previous researches. From our point of view, we think this model fits in the real application scenario of sustainable crowdsourcing. The research result not only has practical guidance to improve enterprise crowdsourcing performance, but can also extent the theoretical of the principal agent model. Given above, we performed some modifications as below:
- In Introduction part, we add some content to illustrate the purpose of enterprise sustainable crowdsourcing, the purpose of this research and the novelty of the work in line#77-90
- In Literature review part, in order to explain why we are using this model, we add an independent paragraph in line#194 to line#207 to list a few previous researches on the quantitative approaches. We also add some words to justify the choice of the model we used. With this information added, we hope it’s more understandable in this research framework.
- In Conclusion part, we emphasized again the practical application of the work, and we also add some advantage and limitation of the model we used, in order to form a rigorous conclusion.
Again, we DO appreciate your feedback, which is very important to improve our research framework, it will help us to further research in the future!
3) Discussion with the results of existing studies - there is currently no reference to the results of other authors' studies referring to the authors' results.
==> This is really good point and we totally agree! Based on your suggestion, we add some discussion with existing studies in line#421-424 and line #474-478.
In addition, we also add some references [12, 30, 38] in line#494 to complete the prior researches on solver’s motivations.
Again, we really thank you for your nice inputs! With your professional insights here, we believe our research will be more rigorous and much better compared to original manuscript.
In the end, during such special COVID-19 period, we really wish you and your loved ones are good and healthy!
Take care and THANK YOU SO MUCH!!!
Reviewer 2 Report
The article is interesting, the subject of research is current. However, I have some remarks:
- In the introduction, it is worth clearly defining the purpose of the work, as well as the novelty of the research.
- In section "The literature review" authors describe different key factors to impact crowdsourcing solver’s motivation. It would be worth creating a list of the most popular factors in the light of previous research.
- In the last paragraph of section “Literature review” authors wrote, that this paper uses the Principal-Agent Model. The choice of the model should be justified.
- In the conclusions the sentences: “Prior researches concluded that the solver’s motivation…” should be completed of references.
- In the conclusions or section of discussion it is worth describe both the advantages and disadvantages of used model.
Author Response
Dear Professor:
First of all, thank you very much for your feedback! These are very good suggestions and we fully agree with your points. Here are our responses in terms of your feedback:
(In order to align the line No., suggest you can use “Tracking Changes = All Markup”)
1. In the introduction, it is worth clearly defining the purpose of the work, as well as the novelty of the research.
==> This is good suggestion! We have mentioned the background and our research question, while your point makes sense to us so we also add the purpose and novelty portion in line#89~92.
2. In section "The literature review" authors describe different key factors to impact crowdsourcing solver’s motivation. It would be worth creating a list of the most popular factors in the light of previous research.
=> We like the idea! Prior literature review is not perfect organized, we think you bring good suggestion to summarize the most popular factors in the light of previous research. So instead of creating the list of the factors, we add Table 2 to clearly list and summarize the most popular factors and typical case in reality.
3. In the last paragraph of section “Literature review” authors wrote, that this paper uses the Principal-Agent Model. The choice of the model should be justified.
==> In order to have justify why the model is used, we add one paragraph from line#194 to line#207 to have more clear explanation for current model, which we believe well fit the sustainable crowdsourcing process in management perspective.
4. In the conclusions the sentences: “Prior researches concluded that the solver’s motivation…” should be completed of references.
==> Nice catch!! We do have that information(in mindJ) when we review and study previous literatures, yes, you are right, we should clearly complete the reference in the right place. Based on your good suggestion, we add some references [18,36,44] in line#494
5. In the conclusions or section of discussion it is worth describe both the advantages and disadvantages of used model.
==> Great suggestion! Although we mentioned the solid conclusion and some limitation of our research in the conclusion part, it’s still necessary to describe both the advantages and disadvantages of the model we used. So follow your opinion, we add the advantage as well as the drawback of this model from line#549 to line#555.
In addition, we did more updated to streamline the research manuscript in a better shape. We reviewed the literature again and updated the literature parts by adding some introduction for sustainable crowdsourcing and solver’s continuous participation. Furthermore, we add some discussions comparison with the results of existing studies in line#421-424 and line #474-478.
Again, we really thank you for your nice inputs! With your insights here, we believe our research will be more rigorous and much better compared to original manuscript.
In the end, during such special COVID-19 period, we really wish you and your loved ones are good and healthy!
Take care and THANK YOU SO MUCH!!!
Reviewer 3 Report
The purpose of this study is to design a incentive mechanism to motivate crowdsourcing participators. A concept named 'sustainable crowdsourcing' based on principal-agent model was proposed and analyzed.
Since the term 'sustainable crowdsourcing' in this paper was proposed by the authors, and it is the majority connection with the journal 'Sustainability' and its audiences. The authors should put more efforts on the definition of 'sustainable crowdsourcing' and the differences between the new concept and existing concepts. There should be a sub-section in literature review section to present existing researches related sustainable crowdsourcing. However, in the references list of this paper, there is no any single word related to 'sustain-' and its derivative.
What is the difference between Feller et al.'s Orchestrating sustainable crowdsourcing and yours?
Joseph Feller, Patrick Finnegan, Jeremy Hayes, and Philip O’Reilly (2012). 'Orchestrating' sustainable crowdsourcing: A characterisation of solver brokerages. Journal of Strategic Information Systems, 21(3), pp. 216-232.
Author Response
Dear Professor:
First of all, thank you very much for your feedback! These are very good insights. Based on your suggestion, we add a few references for sustainability crowdsourcing in literature review. The solver’s participation is critical to achieve sustainable crowdsourcing (we also updated a little bit for our definition in Introduction part), there are a few literatures on this research while few work has been done by using game model, so we do think our work which uses principal-agent model to analyze the solver’s incentive in sustainable crowdsourcing is important from theoretical and practical point of view.
Thank you for providing the reference from Feller, et al., this paper studied the effect and role of solver brokerages who can enable organizations to sustainably acquire distant IP from solvers in three process, including knowledge mobility, appropriability and stability. It’s a little different from perspective of solver’s continuous participation, while we think this is also a very good reference so we include in our latest updates. Thank you so much for providing this!
In addition, we did more updates to streamline the research manuscript in a better shape. We reviewed the literature again and updated the content in a good structure. We also add some discussions comparison with the results of existing studies in line#421-424 and line #474-478. Furthermore, we updated conclusion part to add the advantage and disadvantage of the model we used.
Again, we really thank you for your nice inputs! With your insights here, we believe our research will be more rigorous and much better compared to original manuscript.
In the end, during such special COVID-19 period, we really wish you and your loved ones are good and healthy!
Take care and THANK YOU SO MUCH!!!
Round 2
Reviewer 1 Report
Dear authors
Thank you very much for the changes made to the manuscript. I think aricle has gained a lot.
Good luck in your further scientific work and a lot of health!
Reviewer 3 Report
I have no further questions/suggestions.